# The Role of Veterinarians in Managing Community Cats: A Contextualized, Comprehensive Approach for Biodiversity, Public Health, and Animal Welfare

**DOI:** 10.3390/ani13101586

**Published:** 2023-05-09

**Authors:** Octavio P. Luzardo, José Enrique Zaldívar-Laguía, Manuel Zumbado, María del Mar Travieso-Aja

**Affiliations:** 1Biomedical and Health Research Institute (IUIBS), University of Las Palmas de Gran Canaria, 35016 Las Palmas de Gran Canaria, Spain; manuel.zumbado@ulpgc.es; 2Faculty of Veterinary Medicine, University of Las Palmas de Gran Canaria, 35400 Arucas, Spain; 3Abolitionist Association of Veterinarians against Bullfighting and Animal Abuse (AVATMA), 28045 Madrid, Spain; jezaldivarl@gmail.com; 4Independent Researcher, 35412 Arucas, Spain; marimartravieso@gmail.com

**Keywords:** feral cats, free-roaming cats, trap–neuter–return method, TNR, feline colonies, cat management, adoption, zoonoses, animal welfare law

## Abstract

**Simple Summary:**

Homeless cats pose a significant problem in Europe as they tend to congregate in urban areas where they can find food and shelter. Over time, they may then spread out into natural habitats. Animal welfare organizations provide care to these free-roaming cats, but some stakeholders call for capturing or sacrificing them, which is often illegal and ineffective. Spanish veterinarians urge for a sustainable approach to decrease the population of free-roaming cats through trap–neuter–return (TNR) programs and social awareness, rejecting lethal control and removal methods. A comprehensive study is required to evaluate their actual impact, and effective control programs should focus on non-lethal methods such as TNR and adoption. Public education on sterilization and identification is also needed to prevent abandonment.

**Abstract:**

Homeless cats are a major problem in Europe, with hundreds of thousands abandoned every year. While many die, others can adapt to a lifestyle of roaming freely and establish community cat populations that tend to cluster together in groups. These groups of cats are typically found in urban areas that offer food and shelter to the cats. Animal welfare organizations often care for these cats, providing them with food, shelter, and medical attention. Despite this, conflicts can arise due to the presence of free-roaming cats, with some individuals advocating for drastic measures such as trapping and killing the cats to reduce their populations. However, it is essential to note that such methods are frequently illegal, inhumane, and ultimately ineffective in most situations. A thorough assessment of the impact of cats on a particular natural area requires a comprehensive cat census, a detailed study of the species being preyed upon, and an investigation into the prevalence of zoonotic or epizootic diseases. Moreover, veterinary experts assert that the public health risks associated with cats are often overstated. This article aims to provide a nuanced perspective on the impact of cats on biodiversity in natural areas, while also discussing their role in transmitting the main zoonotic diseases identified in European countries in recent years, with a particular focus on Spain. Effective cat control programs should focus on non-lethal methods such as trap–neuter–return (TNR) and adoption. TNR has proven to be the most effective and humane method of controlling the free-roaming cat population, but its effectiveness is influenced by several factors, including adoption programs and public education on responsible pet ownership. According to Spanish veterinarians, sustainable and science-based solutions such as TNR programs are the best way to achieve population control of free-roaming cats. The veterinary profession should raise awareness regarding sterilization, vaccination, and identification of cats and the consequences of abandonment. They oppose lethal control and removal of cats from the environment, which are ineffective and unethical methods. To promote animal welfare, veterinary professionals must collaborate with public administrations to implement long-term, sustainable solutions to the problem of cat overpopulation. Greater social awareness regarding the importance of sterilization and identification to prevent abandonment and reduce the number of free-roaming cats is also needed. Despite the challenges presented by homeless cat populations in Spain and the rest of Europe, there are many reasons for optimism. Animal welfare organizations and veterinary professionals are actively collaborating to develop humane and effective solutions to manage community cats, including programs such as TNR and adoption. Furthermore, these initiatives are gaining momentum and support from emerging laws and regulations, such as the recent Spanish animal welfare law. Through these efforts, we can reduce the number of free-roaming cats and improve their quality of life.

## 1. Introduction

According to the latest available data, there are approximately 113 million pet cats (*Felis silvestris catus*) in European households [1]. Estimates from some animal welfare organizations indicate that there are several hundreds of thousands or millions of abandoned cats in Europe each year [1,2]. The exact number varies by country and region and can be difficult to accurately determine due to the clandestine nature of animal abandonment [3]. The estimates indicate that, in Spain, the number of abandoned cats is around 150,000 annually [2]. These cats, if they survive, roam in urban, rural, and natural areas, living a life of freedom. However, little is known about the number and distribution of these cats across different habitats, especially in certain regions such as the islandic ones where this article originates. Studies suggest that they generally prefer urban environments, where usually they cluster in groups of fewer than 15 cats [4]. They may disturb neighbors by engaging in fights or trespassing into private spaces, such as gardens or yards. Additionally, issues such as spraying urine and lack of cleanliness and sterilization by feeders can contribute to problems in the area. Conservation groups are concerned about their predation on wildlife and impact on vulnerable species, which hinder species recovery plans [5]. It is commonly reported that hunters, authorities, and the tourism industry claim public health risks or negative impressions associated with free-roaming cats in urban or natural areas. While some complaints are valid, many are based on misconceptions, prejudice, or exaggerated scientific studies used to justify controlling a perceived problem. Despite the various reasons for conflicts, the problem of homeless cats is ultimately due to a lack of comprehensive and effective management.

Cats living in the wild are known by different names, such as “stray cats”, “feral cats”, “colony cats”, or even “wild cats” (which is a different species, *Felis silvestris*), although there is no biological difference from pet cats. The only difference is the degree of socialization with humans [6]. These labels often imply a legal status that allows for intervention against them [7]. In this review, we refer to them as community cats or free-roaming cats.

Conflicts with free-roaming cats often lead to calls for authorities to intervene quickly and decisively [8], which can result in extreme measures such as trapping and culling, poisoning, or shooting. These methods are often illegal and inhumane. Alternatively, capture and transfer to shelters are sometimes considered, but these are often a deferred form of lethal control, as most of these cats are unsuitable for adoption due to their low degree of socialization, and the shelters act as a transitory deposit until they are sacrificed (not euthanized, since it affects healthy individuals) [9]. 

This article provides a veterinary perspective on managing free-roaming cats, with a particular focus on the challenges presented by homeless cat populations in Spain. As veterinary professionals, it is important for us to understand the attitudes of conservationists and work collaboratively with them to address cat overpopulation. While it is still common for cats to be sacrificed as a regrettable but essential reality, it is our responsibility as veterinarians to spearhead changes in animal welfare. The article advocates for TNR programs as the only science-based and humane solution to control the free-roaming cat population [10]. Other methods, such as removing cats from the environment or feeding unsterilized community cats, perpetuate overpopulation and its consequences, which is unacceptable. Effective control programs should prioritize non-lethal methods such as TNR and adoption, with the aim of reducing the cat population and consequently mitigating their impact. Additionally, public education on sterilization and identification is necessary to prevent abandonment. A thorough assessment of the impact of cats on a particular natural area requires a comprehensive cat census, a detailed study of the species being preyed upon, and an investigation into the prevalence of zoonotic or epizootic diseases.

## 2. Dangers Attributed to Cats in the Wild

Free-roaming cats are blamed for harming biodiversity by preying on vulnerable species in natural habitats. They are also usually considered a significant risk to public health. However, some issues associated with free-roaming cats are not easily generalized, so it is important to qualify and contextualize these risks.

### 2.1. Effects on Biodiversity

The impact of cats on vulnerable ecosystems is a topic of debate, with some arguing that they represent the main threat to biodiversity [11], while others suggest that anthropogenic threats outweigh that of predation by cats [12]. Cat populations should be evaluated in each territory to assess their impact on biodiversity, as the level of anthropogenic pressure varies from region to region [13]. A comprehensive study is required for this evaluation [14], and estimates of bird and reptile mortality due to cat predation should be based on scientific data rather than speculation [11]. It is important to note that studies conducted in different geographic areas cannot be generalized [13]. Before considering eradication plans for cats, it is important to assess their actual impact on a specific natural area. Eradication plans could even have negative effects on biodiversity conservation since cats mostly prey on other invasive species [5,15,16].

The presence of cats in natural areas does not always mean a high enough population density to pose a real threat to species conservation. Cats are commensal species whose numbers are highly dependent on human populations [17]. Cats are adaptable to different ecosystems, but they tend to thrive in urban and peri-urban areas due to the abundant resources provided by humans [6,18]. Contrary to popular belief, feeding free-roaming cats does not necessarily increase risks to biodiversity [19]. While well-fed cats may still exhibit hunting behavior [20], individual variations and factors such as age and spaying/neutering status can affect their inclination and success at hunting. Older cats tend to hunt less than younger ones. In this regard, TNR programs that reduce turnover and breeding could result in groups of cats that are older and less prone to hunting [19]. Since cats are primarily found near human populations, efforts to control their density should focus on urban and peri-urban areas [21].

Many scientists argue that cats should not be considered a classical invasive species that can saturate natural habitats and displace native species due to their commensalism and dependence on human populations [21]. Cats have been observed in various ecosystems, and, while specific locations such as dumps or areas with rich food sources may allow for high population densities, their population density is generally limited by the availability of sufficient resources in natural spaces [17]. Even the International Union for Conservation of Nature (IUCN), which lists cats as invasive, acknowledges the difficulty of unequivocally demonstrating that cats cause a decline in prey species due to other factors [22].

The interaction between humans and cats has been ongoing for over 12,000 years, benefitting both species, which makes it challenging to evaluate the overall impact of cat predation on commensal species such as rats, mice, or rabbits in terms of evolutionary ecology [16]. As a result, it would be more fitting to describe the relationship between cats, humans, and biodiversity as complex and fluctuating rather than simply negative [21]. While cats may pose a genuine threat to certain species in some regions, it is essential to examine each case individually based on the natural environment rather than making sweeping generalizations.

### 2.2. Public Health Effects: Reality vs. Perception

Free-roaming cats are often seen as a public health risk due to their potential to transmit zoonotic diseases. However, research shows that the prevalence of these diseases varies depending on location and population. To accurately assess public health risks, it is crucial to consult official sources and consider the context. As veterinarians, we are fully competent in dealing with zoonoses and should refer to official sources for information on diseases cats can transmit. Below is a concise overview of the main zoonotic diseases identified in European countries and their relationship with cats, with special focus on Spain. 

Hookworm disease causes iron deficiency and is mostly found in developing countries [22,23], with few official reports of the disease in Europe [23,24]. The risk of zoonotic transmission of hookworm disease from cats to humans in Europe is low [25,26], and most cases in Europe are specific to humans associated with immigrant populations from countries with poor living conditions [27].

*Ascariasis* is a global intestinal parasitic disease which is mainly found in tropical and subtropical areas with poor sanitation, with *Ascaris lumbricoides* being the primary cause in humans. Zoonotic cases are rare [28]. Dogs and cats can potentially cause intestinal disease in humans, but all reported cases have been related to poor sanitary conditions in developing countries or in immigrants from them. Spain’s official figures do not record any recent cases of this disease, and the sporadic cases reported in Europe have been in newly arrived, irregular immigrants, mainly from Africa. *Toxocara cati* is the primary cause of ascariasis in cats. While occasional cases involving cats have been reported worldwide, they have typically been associated with extremely poor sanitary conditions [24].

*Bartonellosis,* also known as “cat scratch disease”, is caused by the bacteria *Bartonella henselae*, primarily transmitted through cats or fleas that live on them [29]. While no official incidence data are available in Europe, as it is not considered a notifiable disease, studies have shown that cats have a seroprevalence rate of approximately 27% for the disease [30]. The incidence rate in humans is 0.07 cases per 100,000 inhabitants in Spain, with approximately 30–35 cases annually in the country [31]. The disease is believed to have a low incidence rate and is not considered a major public health concern in developed countries, as reported cases are usually mild and can be managed with basic medical care at home. Severe cases that require medical intervention are rare [31]. 

*Cryptosporidiosis* is a parasitic disease that is mandatory to report, and official data show that European countries reported approximately 14,000 confirmed cases in 2017, with a rate of 4.4 confirmed cases per 100,000 population [32]. The main route of transmission is from human to human, with a concentration of cases in children aged 1 to 5 years who are infected through recreational waters such as community swimming pools [32,33]. Zoonotic cryptosporidiosis is mostly associated with infections by *Cryptosporidium parvum* in juvenile cattle [33]. Cats or dogs do not play a relevant or particularly concerning role in the transmission of this disease in the context of the EU [34].

*Dermatophytosis* (ringworm) is a skin condition caused by *Microsporum* spp. and *Trichophyton* spp. fungi. *Microsporum canis* is the most common dermatophyte causing the condition [35]. Cats are infrequent carriers [36], but they may play a role in the transmission of ringworm to humans, especially in close-contact situations. There is a potential risk of dermatophytosis from handling kittens, especially for immunocompromised individuals, and taking measures to avoid exposure is recommended [35,37]. However, the most frequent form of transmission of ringworm in humans is from human to human. It should also be noted that, currently, this disease mainly occurs in developing countries, with sporadic occurrences in Europe [19,35].

*Giardiasis* is a diarrheal disease caused by *Giardia intestinalis*, transmitted through the feces of humans and animals. It is notifiable, and, in 2019, the EU reported 18,004 confirmed cases, with the majority occurring in children aged 1 to 4 years at a rate of 5.2 cases per 100,000 population [38]. Cases occur mainly in the form of outbreaks related to contaminated water and poor handling of food [23,24]. Zoonotic giardiasis is mostly linked to a limited number of animal species, including nonhuman primates, equines, rabbits, guinea pigs, chinchillas, and beavers, either through contact or environmental contamination. Cats are not considered significant contributors to the transmission of this disease [39].

*Dipylidiasis* is a parasitic disease caused by *Dipylidium caninum* and transmitted through fleas. It is commonly found in dogs and cats but rare in humans in developed countries and of little public health concern [40]. Transmission to humans from pets is unlikely as it implies direct ingestion of fleas infected with parasite larvae [40]. External parasite control is easier in cats living in semi-organized groups or households than in those without human surveillance.

*Rabies* is a viral disease that affects the central nervous system, transmitted by the saliva of infected animals, particularly unvaccinated dogs. Although it has been eradicated in some countries, it remains a concern in many parts of the world. The preventive measures include vaccinating pets and avoiding contact with wild animals [41]. The disease is caused by up to sixteen *Lyssavirus* viruses, primarily found in bats, and results in about 60,000 global deaths annually [42]. Cats have a limited role in the transmission of rabies in developed countries given the high vaccination rates of domestic animals, including cats. However, community cats may be at risk of contracting the disease if they come into contact with infected animals and are not vaccinated. Hence, it is crucial to implement community cat management programs in developed countries that include rabies vaccination for these animals [43,44].

Finally, *toxoplasmosis* is the most well-known zoonosis associated with cats. Toxoplasmosis is a highly successful parasitic infection, with at least one third of the world’s population estimated to be infected. As veterinarians, we have a responsibility to educate on the disease and its potential risks to humans, especially to pregnant women. Unfortunately, due to a lack of understanding of the actual risks involved, many cats are abandoned out of fear of contracting the disease [45,46]. Toxoplasmosis transmission is wrongly attributed to cats, as they are the only definitive host of the *Toxoplasma gondii* parasite. While infected cats can shed oocysts in their feces for a few weeks, subsequent shedding does not occur. Nevertheless, the cat’s role in transmitting the disease to humans is minor. The fear of cats transmitting toxoplasmosis to humans is unjustified, as it requires the handling of feces with bare hands and bringing them to the mouth for infection to occur. The risk of direct transmission is negligible except in poor hygienic conditions. Reviews on the disease show that assuming cat contact as the origin of infection is erroneous. A European study showed that pregnant women have virtually no risk of toxoplasmosis from contact with cats, with greater risks from eating undercooked meat, contact with soil, and traveling outside certain regions [47]. In addition to the potential risk for pregnant women, several studies have gained widespread attention for suggesting that toxoplasmosis may cause severe neurocognitive or psychiatric disorders in humans, including but not limited to schizophrenia, suicide attempts, personality changes, and learning difficulties [48]. A 2016 study on a birth cohort of 1000 individuals found no evidence supporting the theory linking toxoplasmosis to an increased risk of psychiatric disorder, poor impulse control, personality aberrations, or neurocognitive impairment. Although the authors did not investigate cats’ influence, they mentioned a popular opinion piece called “Your cat is driving you crazy!” and suggested that the theory may have originated from researchers’ desire to find external explanations for the lack of biological causes that explain common mental disorders and processes [49]. In any case, the only notifiable form of the disease is congenital toxoplasmosis. In the EU, 176 cases of congenital toxoplasmosis were confirmed in 2019, with France accounting for 76% of cases due to its active screening of pregnant women. The notification rate for congenital toxoplasmosis in the EU was 5.1 cases per 100,000 live births [50]. However, France’s screening highlights the importance of early detection. Varying levels of surveillance and screening in other countries may lead to underreporting, posing a public health risk if left untreated. Although relatively rare, untreated toxoplasmosis can cause long-term complications, stressing the need for prevention and early intervention.

## 3. Why Lethal Control of Free-Roaming Cat Populations Is Ineffective

Numerous scientific studies have demonstrated that the trap-and-kill method is an ineffective approach for reducing cat populations permanently. While trapping and killing cats may be effective in certain scenarios, such as on small, uninhabited islands, they require high levels of effort, consistent implementation over time, and can be prohibitively expensive. Additionally, such methods are often illegal and inhumane, and therefore are not a viable or sustainable solution for reducing cat populations in densely populated areas. This is due to the “vacuum effect”, a natural phenomenon that occurs when removing cats from an area only leads to an influx of new cats from nearby regions that move in to access the same resources that attracted the original population [14,51,52,53]. The population in an area where cats have been removed will eventually recover and return to its original size or even increase [54]. There are various possible explanations for this phenomenon. One is that culling operations may target dominant individuals, leading to better resource access for the remaining cats and increased survival rates for juveniles. This compensatory response has been observed in several species after low-level culls [54]. Another possible explanation is that, according to one perspective, cat populations exhibit a hierarchical structure in which dominant cats, usually males, may temporarily exclude subordinate cats from specific areas. This dynamic could potentially result in subordinate cats having wider ranges compared to dominant cats [6,55,56]. In lethal control programs, dominant cats are often targeted first, leaving the “floating” individuals to quickly occupy the empty habitat and contribute to the rapid increase in cat numbers [54]. However, according to some authors, the notion that cat trapping is related to group hierarchy is unfounded. Instead, they propose that trapping programs tend to target cats that are most easily caught, such as those that are hungry, familiar with humans, curious, or for other reasons. Whatever the case, recolonization of areas can occur less than 2 days after poisoned cat and fox baits are placed [57].

Eighty-three successful eradication campaigns of cats have been carried out on islets and islands, but only six of them have been on islands of more than 2000 hectares, all of them uninhabited or sparsely populated. The most referenced case is that of Marion Island, South Africa, where eradication was completed after 20 years of using different lethal methods, such as hunting, trapping, and, finally, introducing feline panleukopenia [58]. However, this approach is not feasible in inhabited areas due to the vacuum effect and the risks of spreading pathogens [59]. Based on the average reported effort of 543 ± 341 person-days per 1000 ha of island over a period of 5.2 ± 1.6 years required for complete eradication of cats and validation of the operation’s success [60], the cost of a complete eradication program for cats from an average European island of 1500 sq km could be estimated to exceed EUR 120 million. However, this estimation assumes that the island is uninhabited, whereas the EU has around 2400 inhabited islands. Moreover, this estimate does not consider the significant impact of human intervention on the islands’ ecosystem or the potential social resistance that such a program could face. 

In addition, another issue with the removal of cats can be lack of predation by other introduced species. An excellent example of this can be found in the study of Bergstrom et al., where it was reported that free-roaming cats had top-down control over the rabbit population on Macquarie Island, and their removal caused a significant increase in rabbit numbers. Although the reduction of the rabbit-control agent *Myxomavirus* was suggested as a factor, further analysis showed that the presence or absence of cats was the main driver of rabbit population size. The study confirms the importance of top-down control by cats and emphasizes the need for careful scrutiny in situations with multiple invasive species both before and after management interventions [16].

Advocates of cat control often oppose lethal methods due to potential social resistance and, instead, favor non-lethal trapping and removal programs that may include adoption [8]. While adoption is a valuable approach in certain areas and circumstances, large-scale adoption programs are not feasible due to the vast number of cats born annually in comparison to the limited number of available homes. Consequently, more than 80% of cats are sacrificed to create space in shelters. Additionally, many free-living cats are too unsocial to be adoptable [9], leading to high return and abandonment rates. Moreover, the vacuum effect still takes place even with non-lethal removal methods, rendering these alternative programs ineffective [9,61,62]. Furthermore, although sanctuaries are frequently suggested as a solution, they should be viewed as a supplementary option, as they often make only a minimal contribution to addressing the problem. Well-run sanctuaries have a limited capacity to take in new cats due to the long lifespan of well-cared-for cats causing them to quickly fill up. TNR programs seek to sterilize cats and then return them to their original location, where they are typically identifiable by ear-tipping. Exceptions may include cases where the cat is adoptable or certain circumstances prevent the cat’s return. For instance, cats that have been shown to specialize in preying on wildlife and pose a real conservation threat, or those with medical conditions requiring special care, may not be suitable for return. Other cats, such as cats that cannot be traced back to a safe return location, such as those found in car engines or other unknown locations, may also require sanctuary care. Therefore, well-run sanctuaries should reserve their limited capacity for cats with special circumstances that prevent their safe return, including those with high predatory behavior, chronic medical conditions, or unknown origins. 

It should be noted that, apart from the inefficacy of lethal control methods, eradication methods such as culling or mass removal may also cause harm to owned cats. This is particularly true in rural areas where cats are permitted to roam freely to control pests. Based on research conducted in an Indigenous community in Australia, where up to 40% of cats euthanized during lethal control programs had owners, causing distress, anger, and social conflicts among them [63], it is possible that a similar phenomenon could occur in other regions where cats are kept on a semi-ownership basis, such as rural areas of the Canary Islands, Balearic Islands, and mainland Spain, according to the authors’ local experience. 

To establish effective control programs for free-roaming cat populations, environmental safety, economic cost, and long-term sustainability should be carefully considered. It is crucial to acknowledge the vast number of cats that exist and plan for sustained efforts rather than sporadic ones. Furthermore, control programs should also consider the growing public affection for cats and the rising sensitivity towards animal welfare in society [64].

## 4. The Importance of Proper Implementation of Neutering as an Effective Solution for Cat Control

Sterilizing a free-roaming cat can enhance its quality of life and deter the birth of numerous kittens, which appears to be a reasonable method for minimizing the overall population of cats. Nonetheless, the efficacy of sterilization in reducing the feline population is a complex process reliant on the biological characteristics of free-roaming cat populations. While some sterilization initiatives have been criticized for failing to significantly impact population size, there is a growing recognition that managing a population differs from managing an individual. Populations have distinct traits and dynamics that cannot be predicted solely from an understanding of individual organisms. Attaining population-level objectives necessitates incorporating fundamental aspects of population biology into trap–neuter–return programs. Veterinarians, armed with their technical expertise and scientific method knowledge, can play a crucial role in designing these programs, not only at the clinical level but also in administrative and planning capacities [14,51,65].

### 4.1. Factors Influencing the Effectiveness of the Trap–Neuter–Return Method for Managing Free-Roaming Cat Populations

In a review conducted in 2020, 66 TNR intervention programs were evaluated, and it was found that the most effective approach is to supplement TNR programs with trapping and removal programs. However, the removal should be limited to cats that can be adopted or are severely ill, unrecoverable, or carrying serious infectious diseases, while the rest of the cats should be returned to the location they came from [9]. The review also highlighted the significance of the long-term maintenance of TNR programs for optimal results [63]. Programs that combined TNR and removal, and were maintained for at least 9 years, achieved population reductions of 54% to 100% [53,66,67,68]. The 12-year controlled field experiment in Tel Aviv, Israel, is one of the most comprehensive studies on the effectiveness of TNR in managing cat populations in an urban area. The study found that TNR must be performed continuously and at high intensity to enable population reduction. To enhance management effectiveness and mitigate compensatory effects, the study recommends evaluating an integrated strategy that combines TNR with complementary methods such as regulating vital resources (e.g., feeding and shelter points), euthanasia of severely ill cats, and adoption [69]. These studies show that the long-term reduction in the number of free-roamingi cats is feasible with a well-implemented TNR method.

While neutering and good health programs may increase cats’ lifespans, it can take some time for populations to naturally decline, leading to frustration for public managers who expect immediate results [70]. Therefore, it is essential to have a technical direction of the program that can set expectations and forecast the expected results over time. Furthermore, adoption programs for sociable cats, when feasible, can significantly accelerate the process [69].

The notion that established groups of cats effectively defend their territory and prevent the immigration of new cats has been challenged by recent studies. While intact male cats typically exhibit a loose hierarchical structure with the dominant (usually largest) cat followed by subordinate cats, neutered cats have been found to display reduced territorial aggressiveness and increased acceptance of new arrivals [71]. Hence, there are proponents of sterilization programs that do not involve gonadectomy, who suggest performing only vasectomy or hysterectomy, as it has been found to be highly effective [72]. However, neutering without removing the gonads has the drawback of not preventing cats’ sexual nuisance behavior in neighborhoods, and the surgical technique is more complex. Nevertheless, McCarthy and colleagues (2013) suggested that any veterinarian with minimal training can effectively perform the procedure [72]. Integrating non-spay sterilization with traditional TNR and adoption of friendly cats could be a viable approach for managing groups of cats located further from residential areas. In addition, studies have explored the use of temporary sterility through hormonal implants. Therefore, comprehensive planning of cat population control programs is necessary, considering these groups of cats as interconnected entities and spanning large geographic areas [14].

The effectiveness of population control programs for free-roaming cats is contingent not only on implementing measures for these cats but also on preventing abandonment [63]. It is surprising to note that established groups of well-fed and healthy cats can inadvertently promote cat abandonment. This is because some owners may release their unwanted cats into these established groups of community cats rather than surrender them to high-kill animal shelters [64,68]. It is vital to educate the public about proper pet care, including measures such as mandatory identification of cats and early and increased sterilization, along with policies that aim not to euthanize solely for space or convenience. Additionally, adequate funding should be provided to support these initiatives [68,73]. Such measures can help to reduce abandonment rates and improve the effectiveness of population control programs in a comprehensive planning approach.

### 4.2. Is the TNR Method Humane?

Critics of the TNR method argue that the “R” in TNR stands for re-abandonment, suggesting that capturing and releasing free-roaming cats is cruel [74]. However, this interpretation is incorrect, and it reflects a misunderstanding of the TNR methodology and the nature of free-roaming cats. TNR aims to capture cats that are already adapted to their habitat and sterilize them to control the population. These cats are not socialized and do not come from a home. Therefore, the “R” in TNR stands for return, not release or re-abandonment, because these cats are returned to the location they came from, where they have already been living. Since cats are highly territorial animals, relocating them can create more problems than it solves [75]. Relocation should only be considered in extreme cases, such as when cats pose significant conservation problems in natural areas. In such cases, relocation should be planned and supervised by specialists [10].

Another reason why TNR programs should not be considered as inhumane is that they can improve the quality of life of community cats. Stella et al. (2013) found that TNR programs can reduce stress related to mating and fighting [75]. Additionally, during sterilization interventions, cats are often vaccinated and dewormed, and groups of cats with caretakers usually receive lifelong care [10].

While it is difficult to provide the same level of welfare to community cats as to pet cats, indicators suggest that the welfare of cats in TNR programs is acceptable. Sterilization helps to avoid high early mortality rates of kittens, which is an important welfare indicator [76]. Moreover, euthanasia rates due to serious diseases are usually very low in community cats (less than 1%), and their body condition is generally good. Although they may be thin, they are not emaciated [77,78]. Sterilization can increase nutrient assimilation and fat deposition, partly due to increased food availability in cared-for groups of cats and reduced physical activity [79].

## 5. Position of Spanish Veterinarians on Cat Population Control: A Comprehensive Overview

As veterinary professionals, we are acutely aware of the pressing need to tackle the issue of shelter sacrifice, which still stands as the primary cause of death for cats [80]. The staggering population of free-roaming cats is far beyond the capacities of adoption, which emphasizes the urgency of the situation. While veterinarians have traditionally accepted the sacrifice of healthy individuals as a regrettable but essential reality, it is incumbent upon us, in the 21st century, to spearhead the changes in animal welfare that society demands. As veterinarians, we concur with other stakeholders that concrete steps must be taken to curb the population of free-roaming cats. However, there is a divergence of opinions concerning the most effective approach to achieve this goal. Simply addressing the consequences of the problem, such as removing cats from the environment, is not a sustainable solution. We must seek to implement solutions that address the root cause of the issue in a sustainable manner, even if they may take longer to implement. This is consistent with our scientific training, code of ethics, and professional responsibility [10].

Thanks to the efforts of veterinarians over the last 20–25 years, many countries, including Spain (where the estimated 2019 castration rate for cats was 74%, despite the lack of official data), have stabilized the populations of registered pet cats through sterilization of males and females [81,82]. However, the focus must also be on free-roaming cats as their sterilization rates remain low, leading to high rates of reproduction and replacement [10,69].

The veterinary profession widely supports TNR (or spaying without neutering) as the sole acceptable method to control the free-roaming cat population, as demonstrated by the support of various organizations such as the American Veterinary Medical Association (AVMA), the Association for Veterinary Epidemiology and Preventive Medicine (AVEPM), the American Association of Feline Practitioners (AAFP), and the Spanish association of small animal veterinary specialists (AVEPA), among others [10,83,84]. Numerous studies have proven its effectiveness and adherence to minimum animal welfare standards, making it the only science-based solution. Although other sterilization methods are being studied, they lack consensual support from the entire veterinary community. Feeding unsterilized community cats without veterinary care perpetuates overpopulation and its consequences, which is unacceptable [10].

Lethal control is strongly opposed, and euthanasia is only recommended in extreme cases when individual health reasons justify it, and always under medical prescription [10]. Although lethal control has been employed in certain limited scenarios, such as on small and uninhabited islands, it is generally not viewed as an effective or humane approach for managing cat populations across larger areas, according to most veterinarians.

Collaboration between veterinary collegiate organizations and public administrations is essential in promoting the implementation of TNR programs. In certain instances, TNR programs are even incorporated into the curricula of veterinary schools, allowing students to practice their surgical skills while providing subsidized services to the community. In addition, some veterinarians provide free advice and assistance to animal welfare organizations’ volunteers in order to facilitate access to sterilization and care for community cats.

Veterinarians must also educate the public about proper pet care, as well as raise social awareness regarding the importance of sterilizing all cats (including pet cats) and the consequences of abandonment. Ultimately, as professionals, we have a responsibility to our code of ethics, our profession, and society to ensure that public administrations apply measures that respect animal welfare.

## 6. Protecting Community Cats: Spain’s Groundbreaking Law for Sustainable and Ethical Management

After years of a debate that has proven to be futile, the management model outlined in this article is increasingly supported as the only truly sustainable and viable long-term solution, as well as being ethical. Even municipal regulations and state laws require the management of community cat populations through reproductive control techniques. The most recent example is the approval in Spain of the law 7/2023 for the protection of animal rights and welfare [85]. This advanced law includes several provisions to ensure animal welfare, including those related to sterilization, vaccination, and identification of community and pet cats. Despite high sterilization rates for pet cats, many still have accidental litters before being sterilized. As a result, the law mandates that all pet cats be sterilized before reaching 6 months of age. The goal is to progressively reduce their population while maintaining their protection as companion animals. 

To accomplish this goal for community cats, it is mandatory that all cats be surgically sterilized and identified through microchip registration under the ownership of the competent local administration. It also establishes the functions of the local administration in relation to community cats. These include the management of community cats through management programs that include, at a minimum, aspects such as the promotion of citizen collaboration in caring for community cats, collaboration with animal protection entities duly registered in the Registry of Animal Protection Entities for the implementation and development of programs, health care for community cats that require it, establishment of action protocols for groups of cats in private locations, training and information campaigns for the population, and population control plans for community cats. The various autonomous governments of the country are required to generate framework protocols with the minimum procedures and requirements to serve as a reference for the implementation of feline management programs in municipal terms. These protocols should include aspects such as mapping and census of the cats in the municipal area, sterilization and identification programs, health programs, and protocols for managing neighborhood conflicts, among other things. Likewise, lines of subsidy have been established in favor of local entities for the fulfillment of their obligations regarding community cats.

In summary, the law seeks to guarantee the welfare of community cats through their population control, identification, and sterilization and the implementation of feline management programs by local entities with the support of autonomous communities. Measures have been established to guarantee citizen collaboration, health care for community cats, and management of possible neighborhood conflicts.

## 7. Conclusions

The article discusses the impact of free-roaming cats on biodiversity and public health in Europe, with a focus on Spain. Spanish veterinarians advocate for TNR programs to control the free-roaming cat population, emphasizing responsible pet ownership and sterilization to address the issue of shelter sacrifice and reduce free-roaming cat populations. Despite being called various names, these cats are biologically no different from pet cats. Extreme measures such as culling, poisoning, or shooting are often illegal and inhumane. To assess their true impact on ecosystems, a thorough study is necessary. To control cat density, efforts should target urban and peri-urban areas. Although cats can transmit zoonotic diseases to humans, the risk is generally low in developed countries.

The trap-and-kill method has been deemed ineffective and unsustainable. Instead, successful cat control programs should focus on non-lethal methods such as spay/neuter-and-return programs, adoption, and sanctuaries for special cases. TNR programs combined with trapping and removal activities are the most effective approach to controlling the population. The veterinary community recognizes TNR as the most humane and effective method of controlling the free-roaming cat population. While sterilization may not necessarily prevent abandonment, it is an important measure to reduce the number of free-roaming cats and prevent them from contributing to the overpopulation problem. Additionally, sterilization can decrease certain obnoxious behaviors, such as urine spraying, that may contribute to abandonment.

## Data Availability

Data is contained within the article.

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
