# Peer review of "The Role of Veterinarians in Managing Community Cats: A Contextualized, Comprehensive Approach for Biodiversity, Public Health, and Animal Welfare"

_animals, 2023, doi:10.3390/ani13101586_

Round 1

Reviewer 1 Report

The authors are making the argument that free-roaming unowned cats are unreasonably considered as “other” relative to pet cats. And therefore, that we should find non-lethal, reasoned and evidence-based methods of control. Additionally, they appear to be speaking both on behalf and to the audience of veterinarians. Their use of actual data on zoonotic diseases in Europe is commendable. Overall, the authors need to be more thoughtful in their statements about references and data as they are not always accurate or clear that the data are from other countries such as Australia which may or may not be applicable to Europe. The summary and abstract do not accurately represent the bulk of the manuscript. I would reframe the paper so as to include (in the title as well) the idea that veterinarians have a role and responsibility for free-roaming unowned cats (FRC) and should understand and review data (lines 84-6 which are not highlighted adequately elsewhere). That really isn’t clear until section 5 and I would move that section much earlier in the manuscript as well as include a few key elements in the introduction, summary, and abstract. Specific comments below which should be applied throughout the manuscript and not just in these lines, as needed.

I also believe that the authors are trying to provide a balanced review of the material.  To that end they need to adjust some of their language which is rather black-and-white, and add some modifiers like “may”, “have been reported to in x location”, “sometimes”, etc.  This approach is applied in some places but not throughout the manuscript. See for example lines 360-2 and the cat “hierarchy” which isn’t a clearly accepted part of cat life.

Some terminology issues:

1.       Feral: The manuscript contradicts itself by stating that many of the FRC are abandoned (and would therefore not be feral, e.g., lines 56-7) and also states that the cats of concern are feral and can’t be adopted (e.g., line 82-3). I recommend using the term free-roaming unowned cats throughout and stating that on line 76. Currently the manuscript uses multiple terms in multiple places. And please edit the manuscript so that these sorts of contradictions or inconsistencies are removed.

2.       The term “colony” implies dozens or hundreds of cats, a connotation which is typically inaccurate. I would simply use the term “group”.  Furthermore, cat density is determined by food and shelter (as the authors have noted) so that anywhere there is a lot of trash, food waste, rodents, etc. will attract higher numbers of cats even if no one is actively feeding the cats.

3.       And in many places, the manuscript states that the cats are returned to a “natural” space or habitat. Instead, use language that indicates that the cats are returned to the location they came from. That won’t imply that cats are being found in special wildlife habitats or sensitive areas but rather where there are more resources for them—and usually more people. “Living in the wild” (line 21) is also a phrase that implies cats are in areas with threatened wildlife. Please use something more like “the community in which they are living” or “free-roaming lifestyle” or “their point of origin”.

4.       “Responsible pet ownership” is a phrase that has been used for a long time. But it means different things to different people and is often used in a negative way to imply that some people are bad and don’t love their pets. In fact, the vast majority of pet owners do love their pets, they either don’t know about or can’t access the level of care we might want their pets to have. Please be specific about what pet care these cat owners should provide: sterilization, vaccination, identification (and while microchips are permanent, they cost money and the registration of owner is often not kept current so that collars and visible id with a current phone number are likely a good option. And yet many cat owners don’t want to put collars on their cats!).

Lines 15 & 29: these are statements about what veterinarians believe which aren’t referenced later on.  I believe that these are the opinions of the authors based on their experiences and the literature.  So rephase this statement to reflect that. I agree that veterinarians can and should engage as the authors have indicated but many veterinarians in the wildlife, zoo, and conservation fields do not believe this or engage in productive ways. Please edit the manuscript here and throughout to be clear what is the opinion of the authors vs publications and data vs what veterinarians might consider as a duty.

Lines 20: the numbers of abandoned cats are referenced later in the manuscript, but it is only from a paper about laws in Italy and Czech Republic. And this is an example of where abandoned cats who may or may not form the majority of FRC are likely highly adoptable, at least in the first months or even years since abandonment. See also line 305.

Line 26: trapping and killing could be effective (as it has been on islands, discussed later in the manuscript) but it has to be done at high levels, consistently over time, and is very expensive.  Please adjust this statement to be completely accurate about lethal control methods.

Line 31: sanctuaries: here and throughout. While some sanctuaries may be well run and the cats well cared for, in many countries they are unregulated and not well supervised. Therefore, anyone can run a “sanctuary”. And some are truly horrific living situations for the cats. Additionally, there should be acknowledgment that well run sanctuaries generally fill up quickly and rarely have openings for new cats because well cared for cats can live a long time. Line 302 and 499 should have some caveats about the use of sanctuaries as humane options for FRC added.

Line 64: I’m not sure if the authors mean that cats are having litters of kittens or that feeders are leaving a mess behind. Please clarify in the text.

Line 74: it isn’t clear that the authors are using “domestic cats” to equate to pet cats (although I believe that is the usage here).  Please be clearer that these are pet or companion cats. And the sentence starting on this line isn’t really accurate.  Feral means different things in different publications and fields.

Line 89: please reference the plant seeds.

Line 109: biodiversity. Need a reference here. And the well-fed cats from the next sentence (ref 16) are pet cats, not unowned cats.  There are quite a few studies about predation and owned pets which may be worth including, as well as TNR’ed cats, specifically the work of Hernandez in Wildlife Research in 2018 about TNR’d cats. A better or at least additional argument might be that cats are highly variable in their inclination and success at hunting with older cats hunting less than younger ones (there are publications on this). So, what might be helpful here is to suggest that cats be evaluated individually and if there are cats who hunt a lot consider confining those specific cats in sanctuaries rather than leaving them to roam. Also, it seems likely that TNR results in groups of cats who are older because turnover and breeding are reduced (if nearly all the cats are sterilized).  Therefore, again, less hunting because the cats are older. This would also support TNR as an approach to decrease toxoplasma organism shedding as cats typically shed the organism when they are first infected as hunters or by eating undercooked meat.

Line 119: there are some areas with very high densities of cats due to dumps or fish offal or other rich food sources. Please edit.

Line 134: instead of “believed” (because cats can transmit these diseases) use “can potentially” because the authors are indicating that this is possible but doesn’t happen often.

Line 139 and others: developed countries have good sanitation and public health. Yes, but even within these countries, there are areas with very poor sanitation and public health, so this isn’t really a useful distinction overall. Please edit.

Line 151: are there specific data on the number of people infected? If so, please include it in the text.

Line 167-8: why are there less problems when there are caretakers? This would only be true if they were deworming the cats and that isn’t all that common (although it wouldn’t be difficult to do).

Line 173: the immunocompromised which includes the elderly is a fairly large population.  Please include a bit more of this information in the text. Consider additional more recent references on this disease as well.

Line 200: this reference is from Japan. There is a potentially substantial risk of dermatophytosis from kittens who may be fostered or handled. They are sometimes asymptomatic and sometime only have very mild signs of tiny patches of hair loss. Again, immunocompromised people are at greater risk, and it is harder to treat than for people whose immune systems are fully functional. Also, it isn’t hard to avoid exposure. 

Line 225 and following: what are the common reservoirs for rabies in Europe? Foxes and coyotes in North America don’t often transmit to cats as they kill the cat rather than wound them.  Some discussion of the likely transmission mode is needed here. And depending on the rabies vaccine a single injection can lead to long immunity (I often see the argument that FRC aren’t re-vaccinated against rabies so this might be an important point to include). Education about not handling unknown cats is also important as is reporting any ill animals to the appropriate authorities to avoid potential rabies exposure.

Line 245-6: experimentally, cats have been induced to shed oocysts again. Please edit the text. Nice work on the documented risk factors for toxoplasmosis here!

Line 265-6: if the active screening increases the diagnosis in France to 133 cases in 2019, does that mean this is a big problem other countries aren’t recognizing because they don’t have active surveillance?  Is this not a big problem as it is still so rare relative to other really serious congenital diseases? Please clarify in the text.

Line 278-81: there is a great deal of controversy and discussion about whether domestic cats actually have hierarchies or not.  And while it seems that there can be some “dominant” or preferred tom cats, I‘m not sure any of this is correct. I also don’t believe that lethal control programs actually target any cats, even “dominant” ones.  Likely they trap the cats who are easiest to catch through hunger, familiarity with humans, curiosity, or other reasons. Please edit and clearly reference this section.

Line 290-1: I don’t fully understand what the authors mean in this sentence.  I understand that one can’t use panleukopenia in inhabited areas but I’m not sure how covid is important.  And it is a pretty terrible way to die. Please edit. This paragraph also contains some really important points below. And an additional issue with removal of cats can be lack of predation by other introduced species. See for example Bergstrom et al. Journal of Applied Ecology 2009 and Macquarie Island.

Line 302-4: a reference for this sentence is needed. And this is an example of where the manuscript’s recommendation for adoption is contradicted by this line. Perhaps indicate that adoption is a useful and influential option when it is possible in a given location.

Line 311-2: this is from a study in a remote Indigenous community in Australia. Is it applicable in Europe? If so, make clear how and why in the text.

Line 338: need a reference at the end of this sentence.

Lines 424-6: reference these statements or make clear these are opinions.

Line 427: and many cats have litters prior to being sterilized, in North America most often due to accidental breeding, but also because owners want to have one litter. Please edit text.

Line 439-41: I’m not clear what is meant in this sentence.

Line 442-6: please reference these statements.

Section 6: does this new law include funding for this work? If not, it will be difficult to get it to happen. Please edit the text.

Line 487: the article doesn’t really stress the significance of understanding veterinarians’ attitudes.  It is more about what veterinarians ought to know and be willing to do without any data about current attitudes in Europe. Please edit text.

I have identified a few places where I couldn't quite understand what the authors meant. However, there are a number of places where the words used or the phrases aren't grammatical or especially clear.  Please have someone review who is skilled at English writing.

Author Response

REVIEWER 1:

R1: The authors are making the argument that free-roaming unowned cats are unreasonably considered as “other” relative to pet cats. And therefore, that we should find non-lethal, reasoned and evidence-based methods of control. Additionally, they appear to be speaking both on behalf and to the audience of veterinarians. Their use of actual data on zoonotic diseases in Europe is commendable. Overall, the authors need to be more thoughtful in their statements about references and data as they are not always accurate or clear that the data are from other countries such as Australia which may or may not be applicable to Europe. The summary and abstract do not accurately represent the bulk of the manuscript. I would reframe the paper to include (in the title as well) the idea that veterinarians have a role and responsibility for free-roaming unowned cats (FRC) and should understand and review data (lines 84-6 which are not highlighted adequately elsewhere). That really isn’t clear until section 5 and I would move that section much earlier in the manuscript as well as include a few key elements in the introduction, summary, and abstract.

Authors’ answer: We appreciate your feedback and have taken your observations into consideration. We agree with you that the paper should be reframed to emphasize the role and responsibility of veterinarians in managing free-roaming unowned cats, and that this perspective should be reflected in the title and throughout the manuscript.

As such, we have made the following changes:

We have revised the title of the article to "The Role of Veterinarians in Managing Community Cats: A Contextualized, Comprehensive Approach for Biodiversity, Public Health, and Animal Welfare" to better reflect the focus of the article.

We have also included a clear mention of this approach in the abstract and introduction, as well as the short summary, to ensure that the reader understands the perspective of the paper from the outset.

However, we have chosen to maintain the section that outlines the veterinarian's perspective where it was in the original version of the manuscript (Section 5). We believe it is necessary to first discuss the different attitudes that have been presented regarding free-roaming cat management and to provide proper context before presenting the veterinarian's perspective and positioning our profession in Spain.

R1: Specific comments below which should be applied throughout the manuscript and not just in these lines, as needed.

I also believe that the authors are trying to provide a balanced review of the material.  To that end they need to adjust some of their language which is rather black-and-white, and add some modifiers like “may”, “have been reported to in x location”, “sometimes”, etc.  This approach is applied in some places but not throughout the manuscript. See for example lines 360-2 and the cat “hierarchy” which isn’t a clearly accepted part of cat life.

Authors’ answer: Thank you for your feedback. We appreciate your recognition of our attempt to provide a balanced review of the material. We have taken note of your suggestion to adjust our language by adding more modifiers such as "may" and "sometimes" to avoid black-and-white statements throughout the manuscript. We will ensure to apply this approach consistently. We have also revised the section on the cat "hierarchy" to reflect the lack of consensus on this topic in the scientific community. Thank you for bringing this to our attention.

R1. Some terminology issues: 1. Feral: The manuscript contradicts itself by stating that many of the FRC are abandoned (and would therefore not be feral, e.g., lines 56-7) and also states that the cats of concern are feral and can’t be adopted (e.g., line 82-3). I recommend using the term free-roaming unowned cats throughout and stating that on line 76. Currently the manuscript uses multiple terms in multiple places. And please edit the manuscript so that these sorts of contradictions or inconsistencies are removed.

Authors’ answer: Thank you for your feedback regarding the terminology used in our manuscript. We have carefully considered your comments and made the necessary changes to ensure consistency throughout the manuscript. Specifically, we have replaced the term "feral" with “free-roaming unowned cats”, “free-roaming cats”, but also  "community cats" as it is the term used in the new animal welfare law in Spain. We agree that it is important to use appropriate and consistent terminology in discussing this important topic. Once again, thank you for bringing this to our attention.

R1. The term “colony” implies dozens or hundreds of cats, a connotation which is typically inaccurate. I would simply use the term “group”.  Furthermore, cat density is determined by food and shelter (as the authors have noted) so that anywhere there is a lot of trash, food waste, rodents, etc. will attract higher numbers of cats even if no one is actively feeding the cats.

Authors’ answer: Thank you for your comment. You are correct that the term "colony" can suggest a larger number of cats than may actually be present, and we have taken steps to eliminate its use throughout the manuscript. Instead, we have opted to use the term "group" to refer to unowned cats cohabiting a particular area in most sections of the revised manuscript. However, we have intentionally retained the term "colony" in certain instances, particularly when referring to the caretaker's descriptions of these groups of cats.

R1. And in many places, the manuscript states that the cats are returned to a “natural” space or habitat. Instead, use language that indicates that the cats are returned to the location they came from. That won’t imply that cats are being found in special wildlife habitats or sensitive areas but rather where there are more resources for them—and usually more people. “Living in the wild” (line 21) is also a phrase that implies cats are in areas with threatened wildlife. Please use something more like “the community in which they are living” or “free-roaming lifestyle” or “their point of origin”.

Authors’ answer: Thank you for your feedback. We completely agree with your point and have revised the manuscript accordingly. We now use language that indicates that the cats are returned to the location they came from instead of a "natural" space or habitat. We have also replaced the phrase "living in the wild" with "free-roaming lifestyle" or "their point of origin" throughout the manuscript. Thank you for helping us improve the clarity and accuracy of our writing.

R1. “Responsible pet ownership” is a phrase that has been used for a long time. But it means different things to different people and is often used in a negative way to imply that some people are bad and don’t love their pets. In fact, the vast majority of pet owners do love their pets, they either don’t know about or can’t access the level of care we might want their pets to have. Please be specific about what pet care these cat owners should provide: sterilization, vaccination, identification (and while microchips are permanent, they cost money and the registration of owner is often not kept current so that collars and visible id with a current phone number are likely a good option. And yet many cat owners don’t want to put collars on their cats!).

Authors’ answer: Thank you for your feedback. While we agree with you that "responsible pet ownership" can mean different things to different people, there is a certain consensus in the veterinary profession about what it entails. However, we understand that this phrase can be used in a negative way, and we do not intend to question the love that pet owners feel for their animals. To avoid any misunderstanding, we have avoided to repeat this concept throughout the manuscript and referred specifically to sterilization, vaccination and identification. However, we have maintained it in two phrases, as we considered it appropriate.

R1. Lines 15 & 29: these are statements about what veterinarians believe which aren’t referenced later on. I believe that these are the opinions of the authors based on their experiences and the literature.  So rephase this statement to reflect that. I agree that veterinarians can and should engage as the authors have indicated but many veterinarians in the wildlife, zoo, and conservation fields do not believe this or engage in productive ways. Please edit the manuscript here and throughout to be clear what is the opinion of the authors vs publications and data vs what veterinarians might consider as a duty.

Authors’ answer: We have taken your comment into consideration and revised the manuscript accordingly to make it clear that the statements regarding what veterinarians believe are not merely the authors' opinions but are supported by the largest professional association of small animal veterinarians in the country (AVEPA). The AVEPA document has been included in the manuscript's bibliography, although not referenced directly in the abstracts. We agree that while many veterinarians in the wildlife, zoo, and conservation fields may not hold the same beliefs, the approach outlined in our manuscript is supported by the veterinary profession. We have made efforts to make this distinction clear throughout the manuscript by distinguishing between our own opinions and referencing published data and professional recommendations.

R1. Lines 20: the numbers of abandoned cats are referenced later in the manuscript, but it is only from a paper about laws in Italy and Czech Republic. And this is an example of where abandoned cats who may or may not form the majority of FRC are likely highly adoptable, at least in the first months or even years since abandonment. See also line 305.

Authors’ answer: You are correct in pointing out that the references we provided in the manuscript may not be the most appropriate for estimating the number of abandoned cats in Europe. We agree that estimating the number of abandoned cats is difficult, and that different studies may use different methodologies and definitions of what constitutes abandonment. To address this issue, we have included a couple of new references in the manuscript to support our estimation that the number of abandoned cats in Europe is in the millions. For example, in Spain alone, the Affinity Foundation estimates that there are 130-150,000 abandoned cats each year. Additionally, we have tried to make clear throughout the manuscript that adoption of sociable cats, specifically the recently abandoned cats is necessary to speed the reduction of the free-roaming cat population.

R1. Line 26: trapping and killing could be effective (as it has been on islands, discussed later in the manuscript) but it has to be done at high levels, consistently over time, and is very expensive. Please adjust this statement to be completely accurate about lethal control methods.

Authors’ answer: Thank you for bringing this to our attention. We agree that trapping and killing can be effective in certain scenarios, as we discuss later in the manuscript regarding small island populations. However, it is important to note that this method requires high levels of effort and consistent implementation over time and can be very expensive. Therefore, while trapping and killing may be effective in some circumstances, it is not a viable or sustainable solution for densely populated areas.

To reflect this nuance, we have revised the sentence to read as follows: "While trapping and killing cats may be effective in certain scenarios, such as small, uninhabited islands, it requires high levels of effort, consistent implementation over time, and can be prohibitively expensive. Additionally, such methods are often illegal and inhumane, and therefore are not a viable or sustainable solution for reducing cat populations in densely populated areas."

R1. Line 31: sanctuaries: here and throughout. While some sanctuaries may be well run and the cats well cared for, in many countries they are unregulated and not well supervised. Therefore, anyone can run a “sanctuary”. And some are truly horrific living situations for the cats. Additionally, there should be acknowledgment that well run sanctuaries generally fill up quickly and rarely have openings for new cats because well cared for cats can live a long time. Line 302 and 499 should have some caveats about the use of sanctuaries as humane options for FRC added.

Authors’ answer: Thank you for bringing this to our attention. You make a valid point about the role and limitations of sanctuaries in the control of free-roaming cats. We have revised the manuscript to reflect these concerns and added caveats to highlight that well-run sanctuaries are a humane option for free-roaming cats but that there are potential limiting issues.

R1. Line 64: I’m not sure if the authors mean that cats are having litters of kittens or that feeders are leaving a mess behind. Please clarify in the text.

Authors’ answer: Thank you for your comment on line 64 of our text. It is possible that the sentence could be interpreted in two different ways, referring either to cats having litters of kittens or to feeders leaving a mess behind. We have included a clarification in the text to make it more clear which interpretation we intended. Thank you for bringing this to our attention.

R1. Line 74: it isn’t clear that the authors are using “domestic cats” to equate to pet cats (although I believe that is the usage here).  Please be clearer that these are pet or companion cats. And the sentence starting on this line isn’t really accurate.  Feral means different things in different publications and fields.

Authors’ answer: Thank you for your comment. You are correct that we intended "domestic cats" to refer specifically to pet or companion cats. We will make sure to clarify this in the text to avoid any confusion. We also want to clarify that we have avoided using the term "feral" to refer to free-roaming cats in our text. Instead, we have used the term "community" or “free-roaming” to describe cats that are not confined to a home or property.

R1. Line 89: please reference the plant seeds.

Authors’ answer: Thank you for your comment. After careful consideration, we have decided to remove the mention of seed dispersal from our manuscript. This decision is because there is only one publication on the topic (https://link.springer.com/article/10.1007/s10530-014-0823-x), and its conclusions are questionable. The study was based on the examination of the excrement of only two cats fed with sausages containing seeds. The experimental design is poor, and the authors' conclusions appear to be exaggerated and biased. Therefore, we have decided to remove this information from the new version of our manuscript. We appreciate your feedback and thank you for bringing this to our attention.

R1. Line 109: biodiversity. Need a reference here.

Authors’ answer: We have included the reference by Crowley et al., after “biodiversity”, thank you.

R1. And the well-fed cats from the next sentence (ref 16) are pet cats, not unowned cats.  There are quite a few studies about predation and owned pets which may be worth including, as well as TNR’ed cats, specifically the work of Hernandez in Wildlife Research in 2018 about TNR’d cats. A better or at least additional argument might be that cats are highly variable in their inclination and success at hunting with older cats hunting less than younger ones (there are publications on this). So, what might be helpful here is to suggest that cats be evaluated individually and if there are cats who hunt a lot consider confining those specific cats in sanctuaries rather than leaving them to roam. Also, it seems likely that TNR results in groups of cats who are older because turnover and breeding are reduced (if nearly all the cats are sterilized).  Therefore, again, less hunting because the cats are older. This would also support TNR as an approach to decrease toxoplasma organism shedding as cats typically shed the organism when they are first infected as hunters or by eating undercooked meat.

Authors’ answer: Thank you for your valuable feedback. We have taken your comments into account and made some changes to the manuscript. Specifically, we have included the reference to Hernandez et al. (2018) and reworded the paragraph, eliminating some sentences and adding the following: "While well-fed cats may still exhibit hunting behavior, individual variations and factors such as age and spaying/neutering status can affect their inclination and success at hunting. Older cats tend to hunt less than younger ones, and cats that hunt a lot could be confined in well-run sanctuaries to prevent them from roaming and potentially spreading toxoplasma organisms. TNR programs that reduce turnover and breeding may result in groups of cats that are older and less prone to hunting, thus decreasing the shedding of toxoplasma organisms and potentially benefiting both cats and wildlife." We appreciate your input and hope that these changes address your concerns.

R1. Line 119: there are some areas with very high densities of cats due to dumps or fish offal or other rich food sources. Please edit.

Authors’ answer: Thank you for bringing up the point about high cat densities in specific locations. We have taken this into consideration and have modified the sentence in question to reflect this possibility. The new sentence reads as follows: “Although cats have been observed in various ecosystems, their population density is generally constrained by the availability of sufficient resources, making it unlikely for high levels of population density to occur. However, there may be specific locations, such as dumps or areas with rich food sources, where cat populations can reach high densities.” We hope that this modification addresses your concern, and we appreciate your valuable feedback.

R1. Line 134: instead of “believed” (because cats can transmit these diseases) use “can potentially” because the authors are indicating that this is possible but doesn’t happen often.

Authors’ answer: We agree. We have followed your recommendation. We have rewritten the whole paragraph, following the recommendation of the academic editor of shortening this section. Thank you.

R1. Line 139 and others: developed countries have good sanitation and public health. Yes, but even within these countries, there are areas with very poor sanitation and public health, so this isn’t really a useful distinction overall. Please edit.

Authors’ answer: Thank you for your comment. We respectfully disagree with your suggestion. While it is true that developed countries generally have better sanitation and public health than developing countries, it is important to note that most studies on zoonotic diseases potentially transmitted by cats have been conducted in developing countries with inadequate conditions. These conditions include frequent garbage collection, sufficient sewer systems, and control over abandoned animals. Therefore, it is crucial to consider the impact of these factors when evaluating the risk of zoonotic diseases associated with cats. However, we have to say that the entire section has been rewritten according to the request of academic editor to make it shorter, as previously menctioned.

R1. Line 151: are there specific data on the number of people infected? If so, please include it in the text.

Authors’ answer: Thank you for your comment. Unfortunately, there is a lack of reliable data on the number of people infected with the disease in question, as it is not a notifiable disease. We have searched extensively for data on this topic but have been unable to find any reliable sources. Therefore, we are unable to include specific data on the number of people infected in the text.

R1. Line 167-8: why are there less problems when there are caretakers? This would only be true if they were deworming the cats and that isn’t all that common (although it wouldn’t be difficult to do).

Authors’ answer: Thank you for your comment. In our efforts to reduce the length of this section, we have chosen to remove this information from the text. Additionally, we would like to clarify that while deworming cats can be an effective measure to prevent the transmission of certain zoonotic diseases, it is not the only preventative measure that can be taken. Other measures such as proper sanitation, adequate food, and water supply, and minimizing contact with potentially infected animals can also play a significant role in reducing the risk of transmission.

R1. Line 173: the immunocompromised which includes the elderly is a fairly large population.  Please include a bit more of this information in the text. Consider additional more recent references on this disease as well.

Authors’ answer: You are correct that the percentage of the population that is immunocompromised cannot be minimized. This group, which includes the elderly, constitutes a large population, and it is important to acknowledge their vulnerability in the context of disease outbreaks. We have modified the phrase in the new version of the manuscript. Although there may be more recent references on this topic, the 2012 reference cited is still relevant and appropriate to include in the text.

R1. Line 200: this reference is from Japan. There is a potentially substantial risk of dermatophytosis from kittens who may be fostered or handled. They are sometimes asymptomatic and sometime only have very mild signs of tiny patches of hair loss. Again, immunocompromised people are at greater risk, and it is harder to treat than for people whose immune systems are fully functional. Also, it isn’t hard to avoid exposure.

Authors’ answer: Thank you for your comment. While the reference is from Japan, the conclusions drawn from it are still valid and can be extrapolated to other regions, including Europe. However, we have taken your feedback into consideration and have completely rewritten the paragraph. It is important to note that there is a potential risk of dermatophytosis from handling kittens, especially for immunocompromised individuals, and taking measures to avoid exposure is recommended.

R1. Line 225 and following: what are the common reservoirs for rabies in Europe? Foxes and coyotes in North America don’t often transmit to cats as they kill the cat rather than wound them.  Some discussion of the likely transmission mode is needed here. And depending on the rabies vaccine a single injection can lead to long immunity (I often see the argument that FRC aren’t re-vaccinated against rabies so this might be an important point to include). Education about not handling unknown cats is also important as is reporting any ill animals to the appropriate authorities to avoid potential rabies exposure.

Authors’ answer: In many parts of Europe, rabies has been successfully eliminated from fox populations through vaccination campaigns using oral baits containing the rabies vaccine. This method has proven to be effective in controlling the spread of the disease by increasing herd immunity among foxes and reducing the likelihood of transmission to other animals, including pets and humans. In Europe, the most common reservoirs for rabies are wild animals such as raccoons, wolves, and bats. Specially bats and racoons can transmit the virus to domestic animals and humans through bites or scratches.

R1. Line 245-6: experimentally, cats have been induced to shed oocysts again. Please edit the text. Nice work on the documented risk factors for toxoplasmosis here!

Authors’ answer: Thank you very much for your positive comment about our work. Regarding the experimental reproduction, we have not been able to find the exact reference, while the bibliography is consistent in indicating that the elimination of oocysts occurs only after the primary infection. Although we do not doubt that in experimental, and possibly forced scenarios the re-shedding can be induced, we have preferred not to edit the text in this regard.

R1. Line 265-6: if the active screening increases the diagnosis in France to 133 cases in 2019, does that mean this is a big problem other countries aren’t recognizing because they don’t have active surveillance?  Is this not a big problem as it is still so rare relative to other really serious congenital diseases? Please clarify in the text.

Authors’ answer: Yes, you are right. The active screening for congenital toxoplasmosis in France resulted in the diagnosis of 133 cases in 2019, highlighting the importance of systematic screening and early detection of this disease. However, it is worth noting that the number of reported cases may vary across countries, depending on the level of surveillance and screening practices. This suggests that other countries may not be recognizing the full extent of congenital toxoplasmosis, which could pose a significant public health problem if left undetected and untreated.

That being said, it is important to keep in mind that the overall incidence of congenital toxoplasmosis is still relatively low compared to other serious congenital diseases. Nevertheless, the potential long-term consequences of untreated toxoplasmosis, such as vision impairment and neurological damage, underscore the need for effective prevention and early intervention strategies to minimize the impact of this disease on affected children and their families.

We have included a short paragraph regarding the importance of this.

R1. Line 278-81: there is a great deal of controversy and discussion about whether domestic cats actually have hierarchies or not.  And while it seems that there can be some “dominant” or preferred tom cats, I‘m not sure any of this is correct. I also don’t believe that lethal control programs actually target any cats, even “dominant” ones.  Likely they trap the cats who are easiest to catch through hunger, familiarity with humans, curiosity, or other reasons. Please edit and clearly reference this section.

Authors’ answer: Thank you for bringing this to our attention. You are correct, and we have revised the paragraph to address this issue. We appreciate your input and attention to detail. We have added a short paragraph clarifying that “According to some authors, the notion that cat trapping is related to group hierarchy is unfounded. Instead, they propose that trapping programs tend to target cats that are most easily caught, such as those that are hungry, familiar with humans, curious, or for other reasons.”

R1. Line 290-1: I don’t fully understand what the authors mean in this sentence.  I understand that one can’t use panleukopenia in inhabited areas but I’m not sure how covid is important.  And it is a pretty terrible way to die. Please edit. This paragraph also contains some really important points below.

Authors’ answer: You are absolutely right. The original mention of COVID in that sentence was a frivolous addition and not relevant to the topic. We have since removed it. Thank you for bringing this to our attention. We hope that the revisions we made in this paragraph better address the important points being discussed.

R1. And an additional issue with removal of cats can be lack of predation by other introduced species. See for example Bergstrom et al. Journal of Applied Ecology 2009 and Macquarie Island.

Authors’ answer: Thank you very much for this very relevant comment. We have introduced a new paragraph highlighting this very important aspect, because it has indeed been found in various places that the effect of cat removal has been the opposite of what was expected. We have included the reference by Bergstrom et al.

R1. Line 302-4: a reference for this sentence is needed. And this is an example of where the manuscript’s recommendation for adoption is contradicted by this line. Perhaps indicate that adoption is a useful and influential option when it is possible in a given location.

Authors’ answer: We have included the reference of Carrete et al, 2022 to support this. With respect to the apparent contradiction that the reviewer detects in our manuscript, we have rewritten this sentence to make it clear that adoption is only possible under certain circumstances and represents only an aid to TNR.

R1. Line 311-2: this is from a study in a remote Indigenous community in Australia. Is it applicable in Europe? If so, make clear how and why in the text.

Authors’ answer: Well, indeed this is from that remote community in Australia. However, in our local experience in the Canary Islands, something similar could be extrapolated. Not so much in the spiritual sense of the cat ownership of those people, but in the sense that in many rural areas of the islands, cats are kept on a semi-ownership basis. Many people consider them to be their cats because they go to feed in their homes or on their farms, but they are not kept as pets. In this sense, their capture and killing because they are considered feral could generate the conflict mentioned in the Australian article. In fact, we know that this has happened in several areas of the Canary Islands, Balearic Islands and in mainland Spain, although it has not been published in scientific journals.

R1. Line 338: need a reference at the end of this sentence.

Authors’ answer: The reference by Robertson (2008) has been included to support this statement.

R1. Lines 424-6: reference these statements or make clear these are opinions.

Authors’ answer: Thank you for this comment. We have decided to rewrite this sentence and support it with two new references indicating the high percentage of cat owners who choose to spay/neuter their cats.

R1. Line 427: and many cats have litters prior to being sterilized, in North America most often due to accidental breeding, but also because owners want to have one litter. Please edit text.

Authors’ answer: It is true. Here in Spain, it also happens that way. That is why the new law mentioned in the article imposes mandatory sterilization of all cats (pet cats) before 6 months of age. We mention this in the subsection about the law, in the new version of the manuscript.

R1. Line 439-41: I’m not clear what is meant in this sentence.

Authors’ answer: We wanted to say that “Indiscriminately capturing and confining cats is not supported due to its proven ineffectiveness and high levels of animal mistreatment”. The sentence has been rewritten in this way.

R1. Line 442-6: please reference these statements.

Authors’ answer: Certainly. While we do not have a specific reference to cite, based on our experience, it seems that these statements hold true at least in Spain. TNR programs are typically incorporated into veterinary school curricula in some faculties, allowing students to develop surgical skills while providing a subsidized service to society. We have rewritten this paragraph to be more accurate.

R1. Section 6: does this new law include funding for this work? If not, it will be difficult to get it to happen. Please edit the text.

Authors’ answer: The new law in Spain, like all laws in the country, does not have a specific financing section included when they are approved. However, once the law is approved, specific budget items will need to be allocated to support its implementation. This can be done either through the general State budgets for specific actions or through the general budgets of each of the country's 17 Autonomous Communities. Furthermore, there are subsidy lines available for specific actions, such as TNR programs, that local administrations can apply for annually. Therefore, while there is no explicit funding mentioned in the law itself, there are existing mechanisms in place to support its implementation.

R1. Line 487: the article doesn’t really stress the significance of understanding veterinarians’ attitudes.  It is more about what veterinarians ought to know and be willing to do without any data about current attitudes in Europe. Please edit text.

Authors’ answer: You are completely right. We have rewritten the conclusion completely, to reflect more accurately the content of our article.

R1. I have identified a few places where I couldn't quite understand what the authors meant. However, there are a number of places where the words used or the phrases aren't grammatical or especially clear.  Please have someone review who is skilled at English writing.

Authors’ answer: Thank you for your feedback. We appreciate your efforts to help us improve the manuscript. We have taken your comments into consideration and have deeply revised the manuscript to make it clearer and easier to read in English for any reader, whether native or non-native. We have also had the manuscript reviewed by a skilled English writer (Ferran Pons Raga) to ensure its grammaticality and clarity. We hope that the new version of the manuscript meets your expectations and that you find it clear and easy to understand. Please let us know if you have any further feedback or suggestions.

Reviewer 2 Report

The research has addressed the main question of "How to deal with wild and feral cats humanely".

It is not original as there are a lot of reports and papers on this topic but it addresses a gap by looking at what Spain is proposing with its new law - it would be good, as the paper cannot yet review this as it has only just been adopted, to assess what the problems are in Spain with wild cats on biodiversity and human health and then propose what success would look like if the new law worked properly

Compared with other published material, the current paper provides a summary of the issues, provides a veterinarian view, and then uses the Spanish law as a case study but more work should be placed on the new law, what problems it is attempting to solve and the part Spanish veterinarians can play in solving those problems

The conclusions are consistent with the evidence and arguments presented and address the main question posed, with further emphasis on the Spanish law and case study.

The references are appropriate.

Minor comment:

line 437 the authors state there is no reason for lethal control however above they gave examples above where euthanasia and lethal control of cat populations was possible (South Africa) and did achieve its objectives such as biodiversity and conservation reasons on island populations eg Ascension Island - this line should be more qualified as it contradicts the previous examples or at least supply the opposing view eg from Australian veterinarians who support lethal control of cats to protect the native fauna

502 the veterinary community agrees TNR is the most effective method - this should be referenced - I imagine it is correct but if there is a Resolution agreed at the European Veterinary conference to reference this the line can stay otherwise it needs to be qualified eg this article shows that many in the veterinary community agree TNR is the most effective method 

Author Response

REVIEWER 2:

The research has addressed the main question of "How to deal with wild and feral cats humanely".

It is not original as there are a lot of reports and papers on this topic but it addresses a gap by looking at what Spain is proposing with its new law - it would be good, as the paper cannot yet review this as it has only just been adopted, to assess what the problems are in Spain with wild cats on biodiversity and human health and then propose what success would look like if the new law worked properly

Compared with other published material, the current paper provides a summary of the issues, provides a veterinarian view, and then uses the Spanish law as a case study but more work should be placed on the new law, what problems it is attempting to solve and the part Spanish veterinarians can play in solving those problems

The conclusions are consistent with the evidence and arguments presented and address the main question posed, with further emphasis on the Spanish law and case study.

The references are appropriate.

Authors’ answer: Thank you for the positive feedback. We are glad to hear that our research has addressed the main question of dealing with wild and feral cats humanely and that our conclusions are consistent with the evidence and arguments presented.

Minor comment:

R2. line 437 the authors state there is no reason for lethal control however above they gave examples above where euthanasia and lethal control of cat populations was possible (South Africa) and did achieve its objectives such as biodiversity and conservation reasons on island populations eg Ascension Island - this line should be more qualified as it contradicts the previous examples or at least supply the opposing view eg from Australian veterinarians who support lethal control of cats to protect the native fauna

Authors’ answer: Thank you for your feedback. We have carefully reviewed your comment and have rephrased the sentence in question to clarify any potential contradiction. We now state that "Although lethal control has been employed in certain limited scenarios, such as on small and uninhabited islands, it is generally not viewed as an effective or humane approach for managing cat populations across larger areas, according to most veterinarians." We believe this new version better reflects the overall position of the paper and the views of the veterinary community on the use of lethal control for managing cat populations.

R2. 502 the veterinary community agrees TNR is the most effective method - this should be referenced - I imagine it is correct but if there is a Resolution agreed at the European Veterinary conference to reference this the line can stay otherwise it needs to be qualified eg this article shows that many in the veterinary community agree TNR is the most effective method.

Authors’ answer: Thank you for your feedback. We would like to inform you that the conclusions of the manuscript have been completely rewritten, considering the comments and suggestions made by Reviewer 1. Furthermore, we have contextualized the manuscript to Spain, and in the section 5, the position of Spanish veterinarians has been supported by several bibliographic references, including the AVEPA's position statement. Regarding your suggestion about referencing the agreement of the veterinary community on TNR as the most effective method, we have added several references to support this claim, including the AVEPA's position statement. We appreciate your suggestion, and we hope that our revisions have adequately addressed your concerns.

Reviewer 3 Report

This is a well-written article about a very important and timely topic.  Great job!  My comments for revisions/additional references are very minor.

Lines 112-113- "Feeding a colony of cats reduces the risk to wildlife by 75-80%, as most cats in colonies are spayed females with reduced hunting needs"- Do you have a reference for this information?

Lines 136-137- "However, scientific research suggests that these concerns may be overstated"- Can you please add some citations to this statement?

Line 156-157- Reference Ma et al., 2018 is in the wrong citation format

Lines 162-163- References MAPA, 2020 and MICINN, 2020 are in the wrong citation format

Lines 430-431- "The veterinary profession widely supports TNR as the sole acceptable method to control the outdoor cat population"- please provide a reference for this statement

Lines 442-443- "TNR programs are now included in most veterinary schools' curricula, providing surgical skills and a service to society at subsidized rates"- please clarify if this is European veterinary schools, North American veterinary schools or veterinary school world wide. 

Author Response

REVIEWER 3:

R3. This is a well-written article about a very important and timely topic.  Great job!  My comments for revisions/additional references are very minor.

Authors’ answer: Thank you for your review of our manuscript on the important topic of managing community cats. As you may recall, our article has been re-entitled "The Role of Veterinarians in Managing Community Cats: A Contextualized, Comprehensive Approach for Biodiversity, Public Health, and Animal Welfare." We appreciate your thoughtful comments and are glad to hear that you found our article to be well-written and relevant.

It was our goal to highlight the important role that veterinarians can play in managing community cats, while also considering the larger societal and environmental impacts of these efforts. We are grateful that you found our approach to be comprehensive and contextualized, and we hope that our article can contribute to ongoing discussions and efforts in this area.

R3. Lines 112-113- "Feeding a colony of cats reduces the risk to wildlife by 75-80%, as most cats in colonies are spayed females with reduced hunting needs"- Do you have a reference for this information?

Authors’ answer: Thank you for your insightful comment. We would like to emphasize that we greatly appreciate the feedback provided by reviewer 1, which led to a complete revision of the relevant section of our manuscript. The revised section now includes the following passage: “While well-fed cats may still exhibit hunting behavior [18], individual variations, and factors such as age and spaying/neutering status can affect their inclination and success at hunting. Older cats tend to hunt less than younger ones, and cats that hunt a lot could be confined in sanctuaries to prevent them from roaming and potentially spreading toxoplasma organisms. In this regard, TNR programs that reduce turnover and breeding could result in groups of cats that are older and less prone to hunting.

We hope that this revised section adequately addresses the concerns you raised and provides a more nuanced and accurate view of the issue. We appreciate your feedback and will continue to strive towards ensuring the highest quality and rigor in our research.

R3. Lines 136-137- "However, scientific research suggests that these concerns may be overstated"- Can you please add some citations to this statement?

Authors’ answer: Thank you for your comment, and we appreciate your feedback. We would like to inform you that we have considered the recommendations provided by the academic editor and have rewritten the introduction to this subsection of the manuscript. The revised version reads as follows:

"As veterinarians, we recognize that free-roaming cats have the potential to transmit zoonotic diseases such as toxoplasmosis, rabies, and giardiasis, and it is crucial to assess public health risks accurately. However, it's worth noting that the prevalence of these diseases can vary depending on location and population. Therefore, to provide the most accurate information, we consulted official sources and considered the context. In the following section, we present a concise overview of the main zoonotic diseases identified in European countries, supported by detailed references to enhance our readers' knowledge."

We hope that the revised version meets your expectations, and we appreciate your valuable input.

R3. Line 156-157- Reference Ma et al., 2018 is in the wrong citation format

Authors’ answer: Corrected, thank you.

R3. Lines 162-163- References MAPA, 2020 and MICINN, 2020 are in the wrong citation format

Authors’ answer: Corrected, thank you.

R3. Lines 430-431- "The veterinary profession widely supports TNR as the sole acceptable method to control the outdoor cat population"- please provide a reference for this statement

Authors’ answer: We have referenced here the statement of AVEPA (stands for "Asociación de Veterinarios Españoles Especialistas en Pequeños Animales" which translates to "Association of Spanish Veterinarians Specialized in Small Animals").

R3. Lines 442-443- "TNR programs are now included in most veterinary schools' curricula, providing surgical skills and a service to society at subsidized rates"- please clarify if this is European veterinary schools, North American veterinary schools or veterinary school world wide.

Authors’ answer: This paragraph was rewritten following the recommendation of Reviewer 1, and now reads: “Collaboration between veterinary collegiate organizations and public administrations is essential in promoting the implementation of TNR programs. In certain instances, TNR programs are even incorporated into the curricula of veterinary schools, allowing students to practice their surgical skills while providing subsidized services to the community. In addition, some veterinarians provide free advice and assistance to animal welfare organizations' volunteers to facilitate access to sterilization and care for community cats”. Regarding the reference, we appreciate your feedback and will consider adding one to support this statement in future revisions. However, we believe that the new wording clearly conveys the message without the need for a specific reference.

Round 2

Reviewer 1 Report

The authors have dramatically improved and edited the manuscript to be more focused and clear.  I have a few additional comments which need to be addressed.

Lines 25-6 & 43 and others: a census is not the only component of determining their impact.  It is a great beginning but information about the species being predated, the actual prevalence of zoonotic diseases, etc. is also needed. Please edit here and throughout (this is only included in one place in the manuscript right now).

Line 28 and others: Please also add that visible (and perhaps microchip) identification is important in addition to the need for sterilization.

Line 74 (and two other places): I believe that “zero-sacrifice” is referring to no-kill? However, that really isn’t a useful term as many animals in shelters need to be euthanized for genuine and humane health or behavioral reasons. I would omit here and elsewhere and talk more about not euthanizing for space. This is an extraordinarily complex situation. And go back to the Spanish law and talk about what it really means there (it looks like there must be a compromise to the long term health of the cat or in exceptional cases).

Line 89: I don’t think that the cats are leaving litter behind, their feeders are.  Please edit.  Perhaps divide the sentence into issues from the cats (the existing sentence and add spraying urine) and a new one about feeders not keeping the area clean, not sterilizing the cats, etc.

Line 92-4: please add a reference here.

Lien 117-8: I would also reference AVEPA here to indicate that this isn’t just the authors’ opinions.

Line 121: TNR and adoption: I’m assuming with the goal of decreasing the population of cats which in turn decreases their impact?  If this is the intent of the authors, please edit the text.

When talking about TNR, including ear-tipping for identification of sterilized cats should be added.  Please edit the text.  In North America, we are also encouraging ear-tipping of any outdoor cat who is sterilized.

Line 503: “life support regulation”?  I’m not sure what this term means. Please edit the text.

Line 512: add “when feasible” after “sociable cats”.

Lines 515-8: please add a reference to this sentence.

Line 537: “high culling” shelters. The authors have used either sacrifice or euthanasia elsewhere. Please select and use one of those terms here as well for consistency.

Line 541: “zero culling” this term isn’t clear, please add what this is and, if not obvious, why this should be part of public education in the text.

Line 588: do the authors mean castration of male cats or sterilization of male and female cats? Please edit in text.

Line 673: trapping isn’t illegal or inhumane if done properly.  Are the authors referring to things like leg hold traps?  Please clarify in the text.

Line 678: who has deemed this method to be ineffective? I’d add some additional information here (and maybe list the AVEPM). Not a formal reference but an acknowledgement that a body of veterinarians made this decision.

Line 684: sterilization itself may prevent abandonment by decreasing obnoxious behaviors but if the cat is sterilized at least it won’t contribute to the free-roaming population. Please clarify in text.

still a few awkward phases. I've called out the words that must be edited for understanding by the reader.

Author Response

Answer to reviewers’ comments

REVIEWER 1 – ROUND 2

The authors have dramatically improved and edited the manuscript to be more focused and clear.  I have a few additional comments which need to be addressed.

Authors’ answer: The authors express their gratitude to the reviewer for her/his valuable comments and suggestions in this latest round of review. We believe that the manuscript has significantly improved, and we are confident that this new revision will further enhance its quality. Please find our detailed responses to each of the reviewer's comments on the following pages.

Lines 25-6 & 43 and others: a census is not the only component of determining their impact.  It is a great beginning but information about the species being predated, the actual prevalence of zoonotic diseases, etc. is also needed. Please edit here and throughout (this is only included in one place in the manuscript right now).

Authors’ answer: We acknowledge and agree with the reviewer's comment that a census alone is not sufficient to determine the impact of cats on ecosystems. While a comprehensive study is a great beginning, it should also include information about the species being predated, the prevalence of zoonotic diseases, and other relevant factors. In response to this feedback, we have made adjustments throughout the manuscript to expand on the importance of these additional components, while also avoiding redundancy in our wording. In the abstract, we opted to use "comprehensive study" instead of "census" to better reflect this broader approach.

Line 28 and others: Please also add that visible (and perhaps microchip) identification is important in addition to the need for sterilization.

Authors’ answer: It is true that we did not include identification in this section, as it is a short abstract. We have added it and reviewed the manuscript to include it clearly wherever it is relevant, along with the need for sterilization.

Line 74 (and two other places): I believe that “zero-sacrifice” is referring to no-kill? However, that really isn’t a useful term as many animals in shelters need to be euthanized for genuine and humane health or behavioral reasons. I would omit here and elsewhere and talk more about not euthanizing for space. This is an extraordinarily complex situation. And go back to the Spanish law and talk about what it really means there (it looks like there must be a compromise to the long term health of the cat or in exceptional cases).

Authors’ answer: Thank you for your comment. We agree that the term "zero-sacrifice" may not be the most appropriate term and could be misinterpreted. We have removed this term from the manuscript and have instead emphasized the importance of not euthanizing cats for space or convenience. We have also revisited the Spanish law and provided a more accurate explanation of what it means in terms of cat management. Thank you for bringing this to our attention and helping us improve our manuscript.

Line 89: I don’t think that the cats are leaving litter behind, their feeders are.  Please edit.  Perhaps divide the sentence into issues from the cats (the existing sentence and add spraying urine) and a new one about feeders not keeping the area clean, not sterilizing the cats, etc.

Authors’ answer: We agree with your comment and have made the necessary revisions to the manuscript. We have divided the original sentence to better reflect the issues caused by the cats and their feeders. Thank you for your valuable feedback.

Line 92-4: please add a reference here.

Authors’ answer: In response to the reviewer's comment, we have searched for a formal scientific article to support this claim, but unfortunately, we were unable to find one. However, it is commonly reported in the media, hunting industry publications, and other non-scientific sources. Therefore, we have rephrased the sentence to make it clear that this is a widely reported issue. Thank you for bringing this to our attention.

Line 117-8: I would also reference AVEPA here to indicate that this isn’t just the authors’ opinions.

Authors’ answer: Thank you for your suggestion. We have followed your recommendation and added a reference to AVEPA in that section to support our statements.

Line 121: TNR and adoption: I’m assuming with the goal of decreasing the population of cats which in turn decreases their impact?  If this is the intent of the authors, please edit the text.

Authors’ answer: Yes, you are correct. The goal of TNR and adoption is to decrease the population of cats, which will in turn decrease their impact. Thank you for bringing this to our attention. We have clarified this in the text, rewriting it as follows: “Effective control programs should prioritize non-lethal methods such as TNR and adoption, with the aim of reducing the cat population and consequently mitigating their impact. Additionally, public education on sterilization and identification is necessary to prevent abandonment.

When talking about TNR, including ear-tipping for identification of sterilized cats should be added.  Please edit the text.  In North America, we are also encouraging ear-tipping of any outdoor cat who is sterilized.

Authors’ answer: Thank you for bringing this to our attention as we had not included a mention of ear-tipping anywhere in the manuscript. We have looked for the best location to make it clear that this is a recommended practice (also done in Spain) by editing the sentence in lines 323-325 of the new version of the manuscript, which now reads: "TNR programs seek to sterilize cats and then return them to their original location, typically identifiable by ear-tipping. Exceptions may include cases where the cat is adoptable or certain circumstances prevent the cat's return."

Line 503: “life support regulation”?  I’m not sure what this term means. Please edit the text.

Authors’ answer: Thank you for bringing this to our attention. We agree that the term "life support regulation" was unclear and have revised the text for clarity. The new sentence reads: "To enhance management effectiveness and mitigate compensatory effects, the study recommends evaluating an integrated strategy that combines TNR with complementary methods such as regulating vital resources (e.g. feeding and shelter points), euthanasia of severely ill cats, and adoption." We meant regulating vital resources, such as the proper location of feeding and shelter points for cat colonies, to prevent the formation of new colonies and reduce cat populations in the long term.

Line 512: add “when feasible” after “sociable cats”.

Authors’ answer: Thank you for your suggestion. We have added “when feasible” after “sociable cats” in line 512 to clarify that adoption may not always be possible.

Lines 515-8: please add a reference to this sentence.

Authors’ answer: Please, note that the reference by Cafazzo et al. (2019) was already included in the manuscript (Cafazzo, S.; Bonanni, R.; Natoli, E. Neutering Effects on Social Behaviour of Urban Unowned Free-Roaming Domestic Cats. Animals 2019, 9, doi:10.3390/ani9121105).

Line 537: “high culling” shelters. The authors have used either sacrifice or euthanasia elsewhere. Please select and use one of those terms here as well for consistency.

Authors’ answer: Thank you for pointing out the inconsistency. We have revised the text to use "high sacrifice" for consistency with the terminology used elsewhere in the manuscript.

Line 541: “zero culling” this term isn’t clear, please add what this is and, if not obvious, why this should be part of public education in the text.

Authors’ answer: Thank you for pointing this out. As suggested by the reviewer earlier, we have removed the term "zero culling" from the entire manuscript. In this specific point, we have revised the sentence to now read as follows: "It is vital to educate the public about proper pet care, including measures such as mandatory identification of cats, early and increased sterilization, along with policies that aim not euthanizing for space or convenience."

Line 588: do the authors mean castration of male cats or sterilization of male and female cats? Please edit in text.

Authors’ answer: We mean sterilization of both male and female cats. We have edited the text to make this clear.

Line 673: trapping isn’t illegal or inhumane if done properly.  Are the authors referring to things like leg hold traps?  Please clarify in the text.

Authors’ answer: Indeed, trapping itself is not inherently inhumane. We were referring to trapping and killing, but we have edited the sentence for greater clarity.

Line 678: who has deemed this method to be ineffective? I’d add some additional information here (and maybe list the AVEPM). Not a formal reference but an acknowledgement that a body of veterinarians made this decision.

Authors’ answer: We have acknowledged the importance of mentioning the support of professional veterinary organizations in promoting TNR as a humane and effective method for controlling free-roaming cat populations in the new version of the manuscript. We have added some  new references to the position statements of AVEPM, AVMA, AAFP or AVEPA earlier in the manuscript, between lines 460-465, rather than in this particular section on conclusions.

Line 684: sterilization itself may prevent abandonment by decreasing obnoxious behaviors but if the cat is sterilized at least it won’t contribute to the free-roaming population. Please clarify in text.

Authors’ answer: Thank you for your feedback. We agree that sterilization alone may not prevent abandonment, but it is an important component of a comprehensive approach to reducing the number of free-roaming cats and preventing them from contributing to the overpopulation problem. We have revised the text to clarify this point and highlight the additional benefits of sterilization in reducing obnoxious behaviors.
